# How Subjective Socioeconomic Status Influences Pro-Environmental Behavior: The Mediating Role of Sense of Control and Life History Strategy

**DOI:** 10.3390/bs14070591

**Published:** 2024-07-11

**Authors:** Bowei Zhong, Nana Niu, Jin Li, Yun Wu, Wei Fan

**Affiliations:** 1CAS Key Laboratory of Behavioral Science, Institute of Psychology, Chinese Academy of Sciences, Beijing 100101, China; zhongbw@psych.ac.cn; 2Department of Psychology, University of Chinese Academy of Sciences, Beijing 100101, China; 3Department of Psychology, Hunan Normal University, Changsha 410081, China; jin.li@hunnu.edu.cn (J.L.); un@hunnu.edu.cn (Y.W.); 4Cognition and Human Behavior Key Laboratory of Hunan Province, Changsha 410081, China; 5Institute of Interdisciplinary Studies, Hunan Normal University, Changsha 410081, China

**Keywords:** socioeconomic status (SES), pro-environmental behavior, sense of control, life history strategy

## Abstract

Understanding the psychological drivers of pro-environmental behavior across different socioeconomic statuses (SESs) is crucial for effectively addressing environmental challenges. To assist businesses and management departments in adequately identifying the psychological characteristics of target consumer groups from different SES backgrounds, our research manipulated subjective SES through three experimental studies to investigate the influence of subjective SES on pro-environmental behavior. Studies 1 and 2 adopted online experiments to examine the influence of subjective SES on pro-environmental behavior within the private sphere and the public sphere. Subsequently, Study 3 further investigated the psychological mechanisms through which subjective SES influences pro-environmental behavior. These results indicated that individuals from high SES backgrounds exhibit a greater propensity for green consumption behavior and contribute more financially to environmental organizations than those from low SES backgrounds. In addition, these studies further elucidated that the sense of control and life history strategy sequentially mediate the relationship. These findings provide empirical evidence for understanding whether and how subjective SES influences pro-environmental behavior, and enriching the theoretical framework of the relationship between subjective SES on pro-environmental behavior.

## 1. Introduction

Pro-environmental behavior refers to the actions taken by individuals that benefit the environment, involving a trade-off between short-term self-interests and long-term environmental benefits [1,2,3,4,5,6], at the expense of self-interests [7,8]. In modern society, rapid economic development often causes human activities that are detrimental to the environment [1], and these have become global threats in the 21st century. Given the adverse effects of self-interested behaviors on the environment, the necessity for environmental protection has escalated in the field of ecological psychology [9].

SES is a complex structure that encompasses both objective material resources, as well as subjective perceptions [10]. Object SES refers to control over tangible or symbolic social resources and is typically measured using indicators, such as wealth, education level, and occupational prestige. Subjective SES is a self-perceived measure of one’s position in the social hierarchy [11,12]. Previous literature has found that the brief priming of subjective SES can elicit behavioral patterns similar to those exhibited by individuals with long-term exposure to a specific SES [13], making it feasible to manipulate subjective SES and study its causal relationship with pro-environmental behavior. Notably, the National Bureau of Statistics released a Gini coefficient of 0.47 for China’s per capita disposable income in 2023, indicating a distinct difference in wealth distribution. Throughout history, society has advocated a cultural value that expects those with abundant resources (i.e., high SES individuals) to contribute more to the overall welfare of society [14]. However, whether individuals of high SES engage more in pro-environmental behavior, as society anticipates, remains inconclusive.

The existing literature provides two research orientations regarding the effect of SES on pro-environmental behavior. On one hand, individuals from high SES backgrounds are more likely to engage in pro-environmental behavior [15]. For example, engaging in green consumption signifies a high social status, with individuals from high SES backgrounds exhibiting higher pro-environmental intentions [16,17]. On the other hand, research suggests that high SES individuals do not necessarily demonstrate greater motivation to support and engage in pro-environmental behavior. For instance, income exhibits a negative correlation with the inclination to participate in pro-environmental behavior, such as increasing the frequency of using public transportation [18]. Additionally, studies have found a weak correlation between pro-environmental behavior and income [19,20,21]. Considering that previous studies have examined either subjective [15,22] or objective SES [20,23] to explore the effect of SES on pro-environmental behavior, the inconsistent results associated with this relationship may originate from different measurement methods. Therefore, the current study employed a comparative approach to prime participants’ subjective SES and explored the causal relationship between subjective SES and pro-environmental behavior. 

Sense of control refers to individuals’ perception of their ability to control events and the degree to which they feel constrained by external factors [24,25], which is the personality variable and is also influenced by the social environment [25,26]. Notably, individuals from low SES backgrounds exposed to resource scarcity experience a diminished sense of control [27,28] and tend to give higher priority to immediate rewards [16,29]. In contrast, individuals from high SES backgrounds experience an elevated sense of control and psychological entitlement [30], which may prioritize long-term benefits. Based on the above findings, considering that pro-environmental behavior involves a trade-off between short-term self-interests and long-term environmental benefits, we discussed the psychological mechanism to examine whether the sense of control might act as a mediator in the relationship between subjective SES and pro-environmental behavior. 

Importantly, existing studies have shown that environmental factors, including financial scarcity (e.g., lacking money), can influence individuals’ life history strategy [31,32]; there are two fundamental life strategies that fall along a continuum ranging from a slower strategy marked by a delayed pace of reproduction and emphasis on offspring quality to a faster strategy characterized by a quicker pace of reproduction and focus on the quantity of offspring [32]. When external cues are predictable and controllable, slower strategies involving delayed reproduction and future investments become adaptive [33]. Conversely, instability, uncertainty, and the severity of the external environment shape individuals’ life history strategies [34], prompting them to follow the fast life history strategy [32], which prioritizes immediate reproduction. Furthermore, the sense of control related to SES may also affect the life history strategy. For example, recent studies have suggested that individuals’ sense of control serves as the psychological motivator linked to the behavioral tendencies associated with both fast and slow life history strategies [33,35]. Based on the above, we posited that individuals from different subjective SESs exhibit distinct senses of control, adopting corresponding behavioral strategies, which may ultimately result in differences in pro-environmental behavior.

Taken together, we conducted three studies using diverse methods to examine whether and why subjective SES influences pro-environmental behavior. Study 1 investigated the causal relationship between subjective SES and the preference for green consumption. Study 2 investigated whether subjective SES influences environmental donation behavior. Study 3 further examined the psychological mechanisms by which subjective SES influences pro-environmental behavior, specifically by investigating the sequential mediation involving the sense of control and life history strategy (See Figure 1). 

## 2. Study 1

Given the prevalence of green consumer behavior in individuals’ daily lives as a form of private-sphere pro-environmental behavior, Study 1 aimed to examine the influence of subjective SES on green consumer behavior. 

### 2.1. Method

#### 2.1.1. Participants

According to G*Power 3.1 [36] with a medium effect size (effect size *d* = 0.5, alpha = 0.05), a sample size of 102 is required to ensure 80% power. To increase the reliability of the results concerning the inconsistent relationship between SES and pro-environmental behavior, we recruited 270 participants from a non-student population using the online platform Credamo [37], a widely used experimental platform that facilitates participant recruitment and data collection. In total, 11 participants who did not follow the instructions (e.g., not following prime imagination of SES) were excluded from the analysis, and 2 exceeded three standard deviations from the mean, thus leaving data from 257 participants for the final analysis. A total of 101 men and 156 women were included (*M_age_* = 29.01 years, *SD* = 8.03). The high SES condition comprised 132 participants, and the low SES condition included 125 participants. Finally, participants were compensated appropriately for their participation.

#### 2.1.2. Procedure and Measurement

Subjective SES manipulation. Participants were shown an image of a 10-rung ladder as a representation of different SESs in China. They were told that the bottom rung represented the lowest SES, with individuals at this level having the fewest resources (lowest income, lowest education level, and least prestigious occupation). Conversely, the top rung represented the highest SES with the most resources (highest income, highest education level, and most prestigious occupation). Importantly, in the high subjective SES priming condition, participants were instructed to compare themselves with individuals of the lowest hierarchy. Conversely, in the low subjective SES priming condition, participants were instructed to compare themselves with individuals of the highest hierarchy [27,38]. In addition, participants were asked to write their feelings when imagining how they would interact with individuals from either the highest or lowest SES, enhancing the priming effects of rank-related states [27,38]. To test the effectiveness of subjective SES, we used the MacArthur Scale [39], in which participants were asked to subjectively choose the position on the ladder that best represented their SES. A higher score indicated a higher subjective SES.

Green consumption preference. The green consumption behavior scale used in this study was based on research by Al Mamun et al. (2018), comprising seven items (e.g., I use energy-efficient appliances) (See Appendix A). Participants were required to rate the extent to which they intended to engage in the aforementioned green consumption behaviors over the upcoming three months using a 7-point Likert scale [40]. A composite measure of green consumption preference was established by averaging the scores for the seven items (α = 0.74), with a higher score indicating heightened levels of green consumption preference.

Objective SES measurement. The objective SES was measured by the family’s annual income, educational level, and occupation (See Appendix A), which was calculated according to previous research [14,41,42]. This involved standardizing and averaging the three indicators of family income, highest educational level, and occupational prestige.

In conclusion, participants completed the subjective SES priming and corresponding manipulation check. They were then assessed for their green consumption preference. Finally, demographic information, including the objective SES, was collected (See Figure 2).

### 2.2. Results

#### 2.2.1. Manipulation Check

An independent samples *t*-test was conducted to examine the manipulation check for subjective SES, which revealed a significant difference between the high (*M* = 5.56, *SD* = 1.06) and low SES priming conditions (*M* = 4.51, *SD* = 1.22, *t* (255) = 7.39, *p* < 0.001, Cohen’s *d* = 0.92), indicating the successful manipulation of the subjective SES.

#### 2.2.2. Green Consumption Behavior

An independent sample *t*-test was conducted to examine the influence of SES on individuals’ preference for green consumption. The results revealed a significant difference between the high (*M* = 5.86, *SD* = 0.60) and low SES priming conditions (*M* = 5.66, *SD* = 0.66) regarding the participants’ green consumption preference (*t* (255) = 2.54, *p* = 0.012, Cohen’s *d* = 0.32), which experimentally supports that individuals with a higher subjective SES have a greater preference for green consumption (See Figure 3). Additionally, when the data within three standard deviations were not excluded, the results still revealed a significant difference between the high (*M* = 5.86, *SD* = 0.60) and low SES priming conditions (*M* = 5.60, *SD* = 0.82), *t* (257) = −2.94, *p* < 0.01, Cohen’s *d* = 0.36.

Furthermore, a regression analysis was conducted with green consumption preference as the dependent variable, subjective SES as the independent variable, and gender, age, and objective SES as the control variables. The results indicated that even after controlling for sex, age, and objective SES, the subjective SES still significantly predicted the participants’ preference for green consumption (*b* = 0.19, *SE* = 0.08, *p* = 0.02) (See Appendix A).

### 2.3. Discussion

Study 1 used experimental manipulation to investigate the impact of subjective SES on the participants’ preference for green consumption, indicating that high subjective SES individuals exhibit a greater preference for green consumption. However, whether these results can be consistently applied to pro-environmental behavior in the public sphere remains unclear. 

## 3. Study 2

Study 2 adopted a real environmental donation paradigm, manipulating participants’ subjective SES and further examining the SES differences in public-sphere pro-environmental behavior (i.e., environmental donation behavior).

### 3.1. Method 

#### 3.1.1. Participants 

We recruited 150 participants through the Credamo online platform. Participants who did not comply with the instructions were excluded, leaving 146 participants for final analysis: 61 men and 85 women (*M_age_* = 28.60 years, *SD* = 7.29). Among these, 73 participants were from the high subjective SES, and the other 73 participants were from the low subjective SES.

#### 3.1.2. Procedure and Measurement

Before the beginning of the experiment, participants were introduced to the World Wildlife Fund (WWF), an organization dedicated to fostering a future in which humans coexist harmoniously with nature. In addition, participants were provided with a CNY 20 participation fee. Subsequently, participants were informed that they had the freedom to donate any amount between CNY 0 and CNY 20 to the WWF. The final participant fee equals the initial participant fee of CNY 20 minus the donated amount [43,44].

After the experiment began, participants first completed the subjective SES priming and manipulation check. Subsequently, they were tasked with deciding what portion of their participation fee they would donate to the WWF. Finally, the objective SES and other demographic information were collected.

### 3.2. Results

#### 3.2.1. Manipulation Check

An independent sample *t*-test was conducted, which revealed a significant difference between the high and low subjective SES conditions (*t* (144) = 5.14, *p* < 0.001, Cohen’s *d* = 0.85). Participants in the high subjective SES (*M* = 5.53, *SD* = 1.07) reported significantly higher ratings than those in the low subjective SES (*M* = 4.59, *SD* = 1.15), suggesting the successful manipulation of the subjective SES.

#### 3.2.2. Amount of Environmental Donation

We conducted an independent sample *t*-test to analyze data on donation amounts. The results indicated a significant difference between participants in the high and low SES conditions (*t* (144) = 4.41, *p* < 0.001, Cohen’s *d* = 0.73). Participants in the high subjective SES condition (*M* = 12.78, *SD* = 5.06) donated significantly more than those in the low subjective SES condition (*M* = 9.17, *SD* = 4.83), further demonstrating consistent SES differences (See Figure 4).

Moreover, a regression analysis was further conducted, with environmental donation behavior as the dependent variable, subjective SES as the independent variable, and objective SES, sex, and age as control variables. The result showed that even when accounting for sex, age, and objective SES, subjective SES continued to be a significant predictor of environmental donation behavior (*b* = 3.36, *SE* = 0.84, *p* < 0.001) (See Appendix A).

### 3.3. Discussion

Study 2 found that high subjective SES individuals donated more to environmental organizations compared to low subjective SES individuals. This study expands on previous findings. Pro-environmental behavior with financial costs typically leads to long-term rewards, which necessitate resistance to immediate gratification [29]. Due to the lack of resources, low SES individuals tend to prefer instant gratification and are less likely to engage in environmentally friendly behavior compared to high SES individuals [16]. Taken together, through these behavioral experiments, we have found the consistent influence of subjective SES on pro-environmental behavior in both the public and private spheres within the cultural context of China, prompting further consideration of the corresponding psychological mechanisms underlying these disparities. 

## 4. Study 3

Study 3 adopted the environmental donation task, aiming to further examine whether individuals’ sense of control and life history strategy may be potential psychological antecedents through which subjective SES influences pro-environmental behavior.

### 4.1. Method

#### 4.1.1. Participants

To investigate the mediating effect, 370 participants were recruited via the online platform Credamo. After excluding 14 participants who did not comply with the instructions and an additional 5 participants with data exceeding three standard deviations from the mean, a total of 351 participants remained. Among these, 128 were men and 223 were women (*M_age_* = 29.08 years, *SD* = 7.53). The low SES prime condition comprised 172 participants, while the high SES prime condition consisted of 179 participants.

#### 4.1.2. Procedure and Measurement

The manipulation of the subjective SES and the subsequent manipulation check followed the procedures used in Study 1. Likewise, the measurement of the objective SES used the same methods as those used in Study 1. Additionally, the procedure used to assess environmental donation behavior was consistent with that of Study 2.

Sense of control scale. Participants filled out the sense of control scale developed by Lachman and Weaver (1998), as revised by Li (2014) [24,33]. The scale comprises two dimensions, namely Personal Mastery and Perceived Constraints (with eight reverse scores), totaling 12 items (e.g., I can achieve almost anything I set my mind to) (See Appendix A). Participants rated each item on a 7-point Likert scale (1 = completely disagree, 7 = completely agree). A higher score indicated a greater sense of control (Cronbach’s α = 0.95). 

Life history strategy scale. The life history strategy scale was evaluated using the Chinese version of the Mini-K scale, which has been translated and revised [45], comprising 20 items (e.g., I often make plans in advance). Participants rated each item on a 7-point Likert scale (1 = completely disagree, 7 = completely agree). A higher score indicated a stronger inclination toward the slow life history strategy, whereas a lower score indicated a stronger inclination toward the fast life history strategy (Cronbach’s α = 0.80). 

In conclusion, participants first completed the subjective SES priming and manipulation check. Subsequently, they filled out the sense of control and life history strategy scales (See Appendix A) and then completed the environmental donation task. Finally, demographic information, including the objective SES, was collected.

### 4.2. Results

#### 4.2.1. Common Method Biases 

Harman’s single-factor test was employed to identify common method bias [46], which involved the variables of subjective SES, objective SES, sense of control, and life history strategy. The results revealed seven factors with eigenvalues >1 before rotation. Additionally, the first factor explained 30.31% of the variance before rotation; below 40% indicated no significant common method bias.

#### 4.2.2. Manipulation Check

An independent sample *t*-test was conducted, which showed a significant difference between the high (*M* = 5.73, *SD* = 1.15) and low SES conditions (*M* = 4.78, *SD* = 1.17, *t* (349) = 7.64, *p* < 0.001, Cohen’s *d* = 0.82), suggesting the effective manipulation of the subjective SES.

#### 4.2.3. Amount of Environmental Donation

A *t*-test was performed on independent samples to confirm the disparities in environmental donation behavior across different SESs. The results revealed that participants in the high subjective SES condition (*M* = 12.22, *SD* = 5.21) donated significantly more to environmental organizations than those in the low subjective SES condition (*M* = 9.66, *SD* = 5.15, *t* (349) = 4.63, *p* < 0.001, Cohen’s *d* = 0.49), further confirming the consistent SES differences in environmental donation behavior. 

Additionally, to control the potential influences of objective SES, sex, and age, a regression analysis was conducted, using environmental donation behavior as the dependent variable, subjective SES as the independent variable, and objective SES, gender, and age as control variables. The results indicated that even with controlling for sex, age, and objective SES, subjective SES remained a significant predictor of environmental donation behavior (*b* = 2.11, *SE* = 0.55, *p* < 0.001) (See Appendix A).

#### 4.2.4. Amount of Sense of Control and Life History Strategy

An independent samples *t*-test was performed to examine the disparities in the respondents’ sense of control and life history strategy. The results revealed that participants in the high subjective SES condition (*M* = 5.13, *SD* = 0.93) had higher sense of control scores compared to those in the low subjective SES condition (*M* = 4.18, *SD* = 1.29), indicating a stronger sense of control (*t* (349) = 7.96, *p* < 0.001, Cohen’s *d* = 0.84). Additionally, participants in the high subjective SES condition (*M* = 5.68, *SD* = 0.46) had higher life history strategy scores compared to those in the low subjective SES condition (*M* = 5.44, *SD* = 0.50), indicating a tendency to adopt slower life history strategies (*t* (349) = 4.71, *p* < 0.001, Cohen’s *d* = 0.50).

#### 4.2.5. Effect of Mediation 

A regression analysis was conducted using the PROCESS macro Model 6 [47], with subjective SES as the independent variable; sex, age, and objective SES as control variables; sense of control and life history strategy as mediating variables; and environmental donation behavior as the dependent variable. Table 1 presents the regression analysis results. The mediation analysis utilized bootstrap resampling (5000 iterations), for which the results are presented in Table 2. These findings indicated that the confidence intervals for the total indirect effect, indirect effect 1, and indirect effect 2 did not include zero, indicating significant mediating roles in the relationship between subjective SES and environmental donation behavior. Specifically, the results suggested that individuals’ sense of control and life history strategy serve as sequential mediators in this relationship. Additionally, the results confirmed that sense of control is an independent mediator of the impact of SES on pro-environmental behavior, but failed to prove that individuals’ life history strategy is an independent mediator in this context.

### 4.3. Discussion

Study 3 further supported the hypothesis that individuals’ sense of control and life history strategy act as mediators in explaining the relationship between subjective SES and pro-environmental behavior. According to social cognition theory, owing to limited access to resources, low SES individuals are inclined toward contextual cognition, experience a lower sense of control over the environment and are more sensitive to external threats [24,27,28]. If individuals perceive themselves as unable to control future outcomes, they will tend to choose immediate rewards and engage in short-term behavior for immediate gains, reflecting the tendency of a fast life history strategy. Conversely, individuals with a stronger sense of control are more likely to prefer long-term benefits, reflecting their preference for a slow life history strategy [33]. These results indicate that individuals’ subjective SES is associated with their sense of control, which in turn predicts their tendency to adopt distinct life history strategies, ultimately influencing the SES differences in pro-environmental behavior.

## 5. General Discussion

Throughout three separate series of studies conducted at various time points, our studies found that compared to low subjective SES individuals, those with high subjective SES backgrounds show a greater preference for green consumerism and donate more money to environmental organization. In addition, these studies further elucidated that the sense of control and life history strategy sequentially mediate the relationship between subjective SES and pro-environmental behavior, revealing the psychological antecedents of this relationship. These findings provide empirical evidence for understanding whether and how subjective SES influences pro-environmental behavior, and providing a potential theoretical basis for promoting pro-environmental behavior among different SESs.

Importantly, consistent findings emerged regarding the influence of SES on pro-environmental behavior in both the private and public domains. These results are supported by previous research. For example, studies have shown a positive correlation between subjective SES and pro-environmental intentions, indicating that individuals from high SES backgrounds have higher pro-environmental intentions [16]. Similarly, pro-environmental behavior is more prevalent among high SES individuals [48,49]. As discussed above, our results indicate distinct pro-environmental behavior patterns among various subjective SESs in the Chinese cultural context.

What is more, our study found that inducing SES affects individuals’ sense of control and life history strategy. These findings are supported by previous research and theories. According to the social cognition theory, individuals from low SES backgrounds typically adopt a contextual cognitive orientation due to limited access to resources and a lower sense of control over the environment, which manifests as feelings of helplessness and a belief that events are uncontrollable [28]. In contrast, high SES individuals are more likely to form self-concepts and social cognitive tendencies based on individual attributes [50], developing a solipsistic cognitive orientation owing to their greater access to resources and higher sense of control over the environment [51]. In addition, life history theory posits that resources are limited, and that all organisms encounter crucial trade-offs in allocating and utilizing their resource “budget” between short-term interests and long-term benefits [31]. According to the life history theory, individuals develop and adjust their strategies for allocating resources to more effectively adapt to the environment [45]. For example, researchers have observed that individuals nearing payday tend to prioritize immediate benefits in intertemporal monetary decisions, indicating higher levels of time discounting [52]. Experience with poverty leads individuals to favor immediate small incomes over potential higher future earnings, resulting in reduced investment in long-term gains [53]. Additionally, experiments manipulating negative income shocks, common among those with a low SES, have shown that such shocks increase delay discounting rates, while positive income changes decrease them [54]. These findings indicate that even a short-term prime of SES can significantly put individuals into a state that is similar to being in that hierarchy for a long time, affecting their sense of control and life history strategies.

Additionally, our study investigated the mediating effects of individuals’ sense of control and life history strategy in the relationship between subjective SES and pro-environmental behavior. The Psychological Shift Model from the perspective of threat highlights three primary ways in which they may influence how individuals behave, namely resource scarcity, environmental instability and unpredictability, and adverse childhood experiences [55,56]. Firstly, individuals control over their life outcomes diminishes, shifting their focus from the future to the present, from distant to local concerns, from distant social relationships to close ones, and from hypothetical to concrete thinking. Secondly, their perceived control weakens, prompting individuals to shift from pursuing long-term goals to short-term ones, from seeking rewards to avoiding threats, and from employing slow strategies to fast ones. The third aspect involves the concentration of cognitive resources on tasks and stimuli to meet immediate urgent needs, leading to adaptive decision-making [56,57]. Based on above, the feeling of scarcity triggered by comparisons with those in the highest hierarchy may reduce individuals’ sense of control and induce psychological shifts in individuals to cope with threats, which may foster a temporary preference for short-term behaviors that provide benefits immediately. Therefore, low subjective SES individuals with a weaker sense of control tend to prioritize immediate rewards, resulting in a reduced consideration of the potential long-term benefits associated with pro-environmental behavior [16], reflecting a fast life history strategy, which prioritizes immediate benefits over engaging in pro-environmental behavior. Conversely, high subjective SES individuals with a strong sense of control lean toward long-term benefits, reflecting a slow life history strategy [33], which tends to engage in pro-environmental behavior that is beneficial for long-term development. Based on these theories and empirical research, these findings indicate that individuals’ sense of control and life history strategy sequentially mediate the influence of subjective SES on pro-environmental behavior, enriching the theoretical framework associated with the relationship between subjective SES and pro-environmental behavior.

The practical implications of this study are twofold. First, it enhances our understanding of how subjective SES influences pro-environmental behavior, providing valuable insights for promoting such behavior across different SESs, which can help businesses and management departments adequately identify the psychological needs of target consumer groups throughout the entire life cycle of green products. Second, considering the distinct psychological and behavioral patterns observed across SESs, efforts to promote pro-environmental behavior should emphasize the long-term benefits of the natural environment and enhance individuals’ sense of control over external events, particularly among low SES individuals.

Our study also has some limitations that warrant further investigation. First, while our study examined the causal connection between subjective SES and pro-environmental behavior and elucidated the corresponding psychological mechanism, a crucial direction for future experimental research involves exploring effective measures to promote pro-environmental behavior across different SESs [9]. Second, our study built upon prior research and categorized SES into high and low hierarchies [50,58,59], unveiling SES disparities within the context of Chinese culture. However, it is noteworthy that SES can also be further stratified into high, middle, and low hierarchies. Future research is warranted to investigate the variations in pro-environmental behavior among individuals from high, middle, and low SES. Third, our study did not find that individuals’ life history strategy mediates the relationship between subjective SES and pro-environmental behavior. Previous studies found that factors such as environmental instability are more likely to influence individuals who grew up poorer, leading them to adopt more short-term behaviors that are consistent with a faster life-history strategy [60]. It is essential to recognize that one’s subjective SES changes over one’s lifetime, resulting in potential differences between one’s childhood and current subjective SES [61], which may be why individuals’ life history strategy failed to mediate the impact of SES on pro-environmental behavior. Therefore, further exploration is needed to investigate the effects of current subjective SES, childhood subjective SES, and individuals’ life history strategy on pro-environmental behavior. Finally, there is a possibility that the smaller donation amounts from individuals with low subjective SES backgrounds may mean that the remaining funds are allocated to other types of pro-environmental behavior post-experiment. Future research could design dedicated studies to more comprehensively rule out this possibility.

## 6. Conclusions

Our study manipulated SES to explore the causal relationship between subjective SES and pro-environmental behavior, highlighting the mediating influence of sense of control and life history strategy. Furthermore, this research lays the foundation for future investigations aimed at devising targeted strategies to enhance pro-environmental behavior among individuals from different socioeconomic backgrounds.

## Figures and Tables

**Figure 1 behavsci-14-00591-f001:**
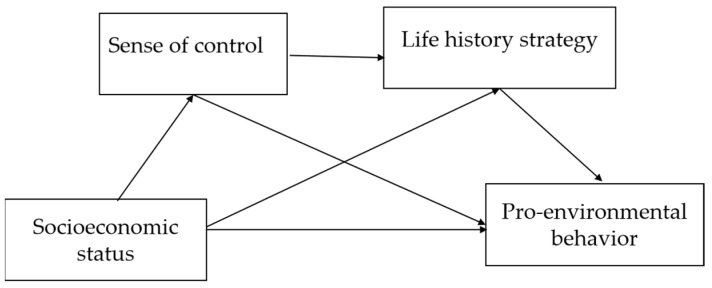
Hypothetical model.

**Figure 2 behavsci-14-00591-f002:**
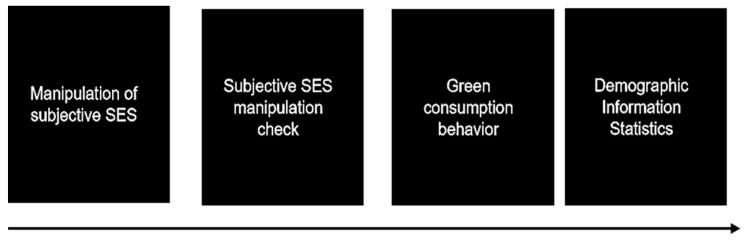
Experimental flowchart.

**Figure 3 behavsci-14-00591-f003:**
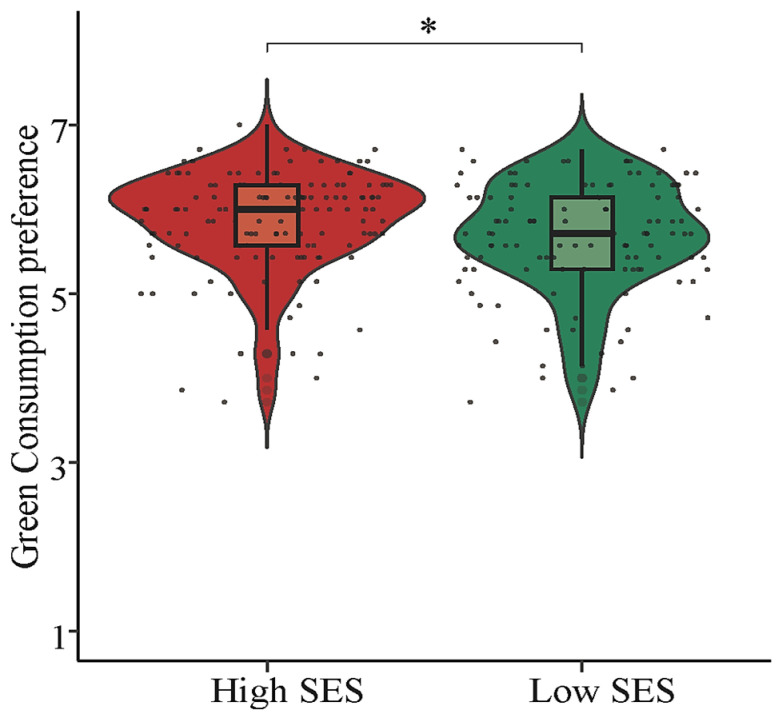
The green consumption preference across different SESs, * *p* < 0.05.

**Figure 4 behavsci-14-00591-f004:**
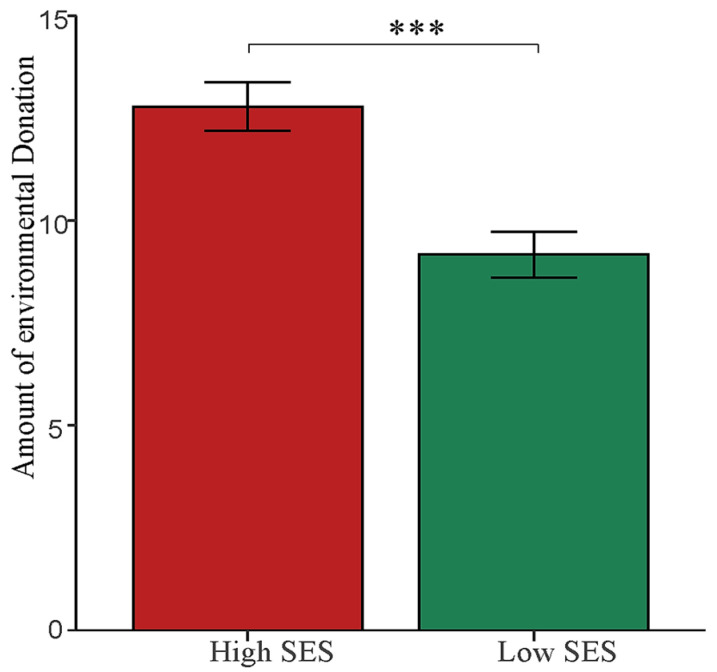
The amount of environmental donation across different SESs. *** *p* < 0.001. Error bars indicate the standard error of the means (SE).

**Table 1 behavsci-14-00591-t001:** Hierarchical multiple linear regression predicting impact of subjective SES, sense of control, and life history strategy on environmental donation behavior.

Variable	Sense of Control				Life History Strategy				Environmental Donation Behavior			
	*b*	*SE*	*t*	CI	*b*	*SE*	*t*	CI	*b*	*SE*	*t*	CI
Intercept	3.80	0.26	14.68 ***	[3.29; 4.31]	4.54	0.12	37.12 ***	[4.30; 4.78]	−5.14	3.25	−1.58	[−11.54; 1.26]
Subjective SES	0.86	0.12	7.28 ***	[0.63; 1.10]	0.02	0.05	0.52	[−0.07; 0.12]	0.92	0.56	1.63	[−0.19; 2.03]
Sense of control					0.23	0.02	11.32 ***	[0.19; 0.27]	1.05	0.28	3.79 ***	[0.51; 1.60]
Life history strategy									1.27	0.64	1.98 *	[0.01; 2.53]
Objective SES	0.33	0.09	3.82 ***	[0.16; 0.50]	0.03	0.03	0.80	[−0.04; 0.09]	0.26	0.39	0.66	[−0.51; 1.03]
Sex	0.06	0.12	0.47	[−0.18; 0.30]	−0.05	0.05	−1.01	[−0.14; 0.04]	1.08	0.54	2.00 *	[0.02; 2.14]
Age	0.01	0.01	1.62	[−0.00; 0.03]	0.00	0.00	−0.15	[−0.01; 0.01]	0.10	0.04	2.85 **	[0.03; 0.17]
*R* ^2^	0.20				0.33				0.19			
*F*	22.06 ***				34.45 ***				13.77 ***			

Notes. b = unstandardized coefficients. CI = percentile bootstrapped confidence intervals. * *p* < 0.05, ** *p* < 0.01, *** *p* < 0.001.

**Table 2 behavsci-14-00591-t002:** Mediating effects of sense of control and life history strategy between subjective SES and environmental donation behavior.

Pathway	Effect	SE	95% CI	
			LL	UL
Total indirect effect	1.189	0.269	0.703	1.746
Indirect effect 1	0.248	0.120	0.018	0.492
Indirect effect 2	0.910	0.272	0.421	1.500
Indirect effect 3	0.031	0.065	–0.089	0.177

Notes. Indirect effect 1 = subjective SES → sense of control → life history strategy → environmental donation behavior. Indirect effect 2 = subjective SES → sense of control → environmental donation behavior. Indirect effect 3 = subjective SES → life history strategy → environmental donation behavior. CI = percentile bootstrapped confidence intervals, LL = lower limit, UL = upper limit.

## Data Availability

The data supporting the findings of this study are available from the corresponding author upon reasonable request.

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
