# Peer review of "How Subjective Socioeconomic Status Influences Pro-Environmental Behavior: The Mediating Role of Sense of Control and Life History Strategy"

_behavsci, 2024, doi:10.3390/bs14070591_

Round 1
Reviewer 1 Report
Comments and Suggestions for Authors
Dear colleagues,
Please, make shorter your "Introduction" part. Why we would make a paper where almost 2 pages are dedicated to introduction? Within the methodology of the scentific paper, introduction is the entrance door towards explanation of the presented hypothesis. Just entrance door. Please, make it shorter, than it can be published.
In regard the rest of the paper, when doing research, number of 150 participants are good, but for the next time make it levelized - i.e. Study 1 is 270, Study 2 is 150 and Study 3. is 370 participants to the survey. That is not good (Looks like it was open possibility and not target one -i.e. not chosen before, but open for comments of anyone who is there). Would be much better, i.e. 300 each. Using like this today is ok, but would be more methodological to increase to proposed number next time and, by all means, choose the groups in regard their income, place of living, education, age, sex, interests...
"Conclusion" itself has to be opposite to "Introduction" part. Namely, first is too short and the second one is too long. Work on it, please. "Conclusion" has to be scientifically focused on outcomes of the research and how it will benefit the society in regard SES future.
Author Response
Responses to the reviewer 1's suggestions
Dear Reviewer:
Thank you very much for your suggestions. We appreciate and cherish the constructive and valuable comments you have given. Now we have done our best to address all comments in our resubmission. The blue words in the manuscript belong to changes or additions. Below are specific responses to your suggestions. We have also provided a PDF file in response to your suggestions.
Comments 1: Please, make shorter your "Introduction" part. Why we would make a paper where almost 2 pages are dedicated to introduction? Within the methodology of the scentific paper, introduction is the entrance door towards explanation of the presented hypothesis. Just entrance door. Please, make it shorter, than it can be published.
Response: Thank you for your comments and valuable suggestions. Based on your suggestions, we have condensed the introduction. Due to its length, we haven't included the revised introduction in this response. We appreciate your expertise once again.
Comments 2: In regard the rest of the paper, when doing research, number of 150 participants are good, but for the next time make it levelized - i.e. Study 1 is 270, Study 2 is 150 and Study 3. is 370 participants to the survey. That is not good (Looks like it was open possibility and not target one -i.e. not chosen before, but open for comments of anyone who is there). Would be much better, i.e. 300 each. Using like this today is ok, but would be more methodological to increase to proposed number next time and, by all means, choose the groups in regard their income, place of living, education, age, sex, interests...
Response: Thank you for your comments and valuable suggestions. Based on previous research, the primary reason for the sample size design in this study is the inconsistent findings regarding the relationship between SES and pro-environmental behavior. Therefore, Study 1 recruited a larger number of participants (270), significantly exceeding the 102 participants required to ensure 80% power according to G*Power 3.1 (medium effect size d = 0.5, alpha = 0.05) (Faul, Erdfelder, Lang, & Buchner, 2007). Study 2, on the other hand, recruited fewer participants based on the results of Study 1 to further examine whether there are differences in the SES effect across different types of pro-environmental behavior. For Study 3, considering that verifying the mediation model requires a larger sample, the sample size was increased again.
We acknowledge that this design rationale is not entirely convincing. We greatly appreciate the expert's insights into the sample size design issues, which will be highly beneficial for our future scientific research. Thank you very much for your valuable suggestions.

Reviewer 2 Report
Comments and Suggestions for Authors
This paper examines the association between subjective SES (the participant's sense of their position on the SES ladder) and their environmental intentions and behaviors. It demonstrates that a simple experimental manipulation of subjective SES can produce differences in both the intention to engage in pro-environmental consumption and in the size of a donation made to an environmental organization. The paper also looks at whether subjective SES is associated with those same outcomes when controlling for other predictors, including objective SES. Finally, the papers looks at whether other constructs, sense of control and life history strategy, mediate the association between subjective SES and donations.
In general, I think the paper does a very good job of demonstrating that this manipulation 1) impacts subjective SES, and 2) impacts expressed intention to engage in pro-environmental behavior and donations to an environmental organization. At some points the authors go a little beyond what they can claim based on this study (they don't demonstrate an effect on pro-environmental consumption, only on participants' expressed intentions and they don't demonstrate that this is due to SES, only due to manipulating subjective SES), but the basic findings are well-evidenced.
I'm a little concerned, though, with interpretations of the regression models and the theoretical motivations. The regression models as currently specified cannot support causal inferences. Subjective SES is impacted by the experimental condition, but it's not assigned at random. People with high subjective SES may differ from people with lower subjective SES in other ways as well. Any results coming from the regression models should be interpreted as associations, not as demonstrating an "influence". If the authors want to examine the causal effect of subjective SES on the outcomes, they should consider using a method like instrumental variables, causal mediation, or principal stratification. Alternately, they should be clear that these estimates are not causal in nature. It's not so much that they explicitly state that they've found evidence of a causal effect of subjective SES on other variables, it's that they don't clearly delineate between the causal estimates and the non-causal. Given that all of these studies involve experimental manipulation, it's very important for the authors to explicitly flag those estimates which do NOT support causal inferences.
Similarly, the rationale for the design of study 3 is not really clear to me. Study 3 examines two questions. First, what is the impact of the intervention on donations. That's already been addressed in study 2, and while there's nothing wrong with replicating the results, there doesn't seem to be a need to do so in the same paper. Second, it looks at a path analysis where donations are related to subjective SES, with sense of control and life history strategy as mediators. This is fine, but is non-causal in nature, and it seems like the intervention is actually making it harder to understand the various paths. For example, it seems unlikely that this intervention could impact life history strategies, which are about how the participant tends to make decisions. But this construct is used as a mediator for the association between subjective SES and donations, both of which ARE impacted by the experimental manipulation. It seems like this research question would be better served by a design which didn't involve the experimental manipulation. Regardless, it's important for the authors to explicitly acknowledge that these path estimates are non-causal and are simply describing associations between variables. That's especially true here where the experimental design and the path model might lead some readers to think that this model is estimating causal effects.
Below find some more detailed notes.
Introduction:
I don't have a strong background in this area, and will not have much to add here. However, I did have a question about the logic in the paragraph beginning on line 56. If SCT predicts that people with different levels of SES will develop different cognitive styles, how would manipulating subjective SES in a one-time experiment impact those? Presumably stable cognitive styles are built up over a lifetime, so why would they be impact by a simple experimental manipulation?
Similarly, I don't really know the theory of life history strategy, but can it really be significantly impacted by an intervention in a laboratory?
Line 43: I believe this should be cited in a conventional format.
Study 1:
Line 164: Why are the researchers looking for a medium effect size?
Line 167: There should be supplemental information about who tends to participate in this online platform. Without knowing who was likely to be in the sample, there's no way of knowing what groups these results will generalize to.
Line 170: The analyses should also be run including the excluded participants (at least the ones with extreme scores) as a robustness check.
Line 223: The claim that this validated the causal relationship of SES and pro-environmental behavior is too strong. The researchers have demonstrated a causal relationship between subjective SES and expressed intention of pro-environmental behavior, but this is a slightly weaker claim.
Line 227: I don't see the need for this regression since the experimental design already rules out sex, age, and objective SES as reasons for the association. There's nothing wrong with including it, though. However, I think the researchers could use an instrumental variables approach to detect the causal effect of manipulating subjective SES on the outcome. Given the way the model is currently specified, I don't see how this can support any causal claims, since subjective SES varies for reasons that are not related to the experiment. The researchers claim that subjective SES "influences" the outcome is not supported by this model.
Study 2:
Similar comments to before. In this case researchers DO demonstrate that manipulating subjective SES affects pro-environmental behavior.
Line 285: This study does not demonstrate that "individuals from high SES donate more money to environmental organizations, whereas those from low SES donate less". Rather, it demonstrates that people induced to have a higher subjective SES tend to donate more.
Line 289: It may be true that "low SES individuals prefer instant gratification and are less willing to engage in environmentally friendly behavior for long-term development", but this study isn't enough to make this claim. In particular, I don't think the authors can claim that people assigned to the low subjective SES group are prioritizing instant gratification, since it's not known what they did with the money they chose not to donate.
Study 3:
I'm curious why the authors didn't use a structural equations model, given that sense of control and life history strategy are both latent variables (and, of course, the outcome). Also, why did this study use the experimental manipulation? Study 2 already demonstrated that the manipulation impacts giving, and the study design does NOT allow the authors to claim that the mediation is causal (since subjective SES varies for reasons unrelated to the intervention), so I'm not clear why the experimental manipulation was used here.
I think the authors should make more of the fact that life history strategy was not related to subjective SES. On the one hand, this is good since an experimental manipulation shouldn't be able to impact life history strategy. On the other hand, subjective SES varies for reasons that are not related to the intervention, and it seems surprising to me that these variables would not be associated, especially given the introduction of the paper where the authors claimed that SES should be related to life history strategies (and not only due to differences in sense of control). At the least this should be discussed.
Discussion
Line 424: I don't see how a simple intervention manipulating subjective SES could be related to "form[ing] self-concepts and social cognitive tendencies based on individual attributes". Similarly, does life history theory allow for changes in strategies related to extremely small interventions like the one used in this study?
Comments on the Quality of English LanguageThere are a few minor errors that should be cleaned up, but none that impact the readability of the paper.
Author Response
Responses to the reviewer 2's suggestions
Dear Reviewer:
Thank you very much for your suggestions. We appreciate and cherish the constructive and valuable comments you have given. Now we have done our best to address all comments in our resubmission. The blue words in the manuscript belong to changes or additions. Below are specific responses to your suggestions. We have also provided a PDF file in response to your suggestions.
This paper examines the association between subjective SES (the participant's sense of their position on the SES ladder) and their environmental intentions and behaviors. It demonstrates that a simple experimental manipulation of subjective SES can produce differences in both the intention to engage in pro-environmental consumption and in the size of a donation made to an environmental organization. The paper also looks at whether subjective SES is associated with those same outcomes when controlling for other predictors, including objective SES. Finally, the papers looks at whether other constructs, sense of control and life history strategy, mediate the association between subjective SES and donations.
Comments 1: In general, I think the paper does a very good job of demonstrating that this manipulation 1) impacts subjective SES, and 2) impacts expressed intention to engage in pro-environmental behavior and donations to an environmental organization. At some points the authors go a little beyond what they can claim based on this study (they don't demonstrate an effect on pro-environmental consumption, only on participants' expressed intentions and they don't demonstrate that this is due to SES, only due to manipulating subjective SES), but the basic findings are well-evidenced.
Response: Thank you for your comments and valuable suggestions. Based on previous research, we have revised "green consumption behavior" to "green consumption preference" to enhance the scientific accuracy and readability for the readers (Chuang, Xie, & Liu, 2016), and also modify the study's description of SES on pro-environmental behavior to subjective SES on pro-environmental behavior.
Comments 2: I'm a little concerned, though, with interpretations of the regression models and the theoretical motivations. The regression models as currently specified cannot support causal inferences. Subjective SES is impacted by the experimental condition, but it's not assigned at random. People with high subjective SES may differ from people with lower subjective SES in other ways as well. Any results coming from the regression models should be interpreted as associations, not as demonstrating an "influence". If the authors want to examine the causal effect of subjective SES on the outcomes, they should consider using a method like instrumental variables, causal mediation, or principal stratification. Alternately, they should be clear that these estimates are not causal in nature. It's not so much that they explicitly state that they've found evidence of a causal effect of subjective SES on other variables, it's that they don't clearly delineate between the causal estimates and the non-causal. Given that all of these studies involve experimental manipulation, it's very important for the authors to explicitly flag those estimates which do NOT support causal inferences.
Response: Thank you for your comments and valuable suggestions. Although our t-test indicated that subjective SES affects pro-environmental behavior, we recognized that the participants in our online experiment were from non-student samples, which might introduce heterogeneity in objective SES between groups. To eliminate the potential confounding effect of objective SES, we conducted a regression analysis.
Your concerns have significantly improved our manuscript. We have altered the presentation of the causal relationship between SES and pro-environmental behavior to emphasize the correlation in the regression analysis. In addition, we have further studied the causal mediation models (Ge, 2023), which have been crucial for informing our future experimental designs on causal mediation models. Thank you once again for your valuable suggestions.
Comments 3: Similarly, the rationale for the design of study 3 is not really clear to me. Study 3 examines two questions. First, what is the impact of the intervention on donations. That's already been addressed in study 2, and while there's nothing wrong with replicating the results, there doesn't seem to be a need to do so in the same paper. Second, it looks at a path analysis where donations are related to subjective SES, with sense of control and life history strategy as mediators. This is fine, but is non-causal in nature, and it seems like the intervention is actually making it harder to understand the various paths. For example, it seems unlikely that this intervention could impact life history strategies, which are about how the participant tends to make decisions. But this construct is used as a mediator for the association between subjective SES and donations, both of which ARE impacted by the experimental manipulation. It seems like this research question would be better served by a design which didn't involve the experimental manipulation. Regardless, it's important for the authors to explicitly acknowledge that these path estimates are non-causal and are simply describing associations between variables. That's especially true here where the experimental design and the path model might lead some readers to think that this model is estimating causal effects.
Response: Thank you for your comments and valuable suggestions.
Regarding the first issue: based on previous research, we continued to use the environmental donation task in Study 3 because it measures actual pro-environmental behavior rather than mere preferences for pro-environmental behavior. Additionally, obtaining consistent results with the same task helps confirm the stability of our findings. However, as you noted, replicating the same results within the same paper may not always be necessary.
Regarding the second issue: we acknowledge that the mediation model results do not imply causality. We have revised our statements to avoid suggesting that the mediation paths indicate causal relationships. Thank you again for your insightful suggestions.
Below find some more detailed notes.
Introduction:
Comments 4: I don't have a strong background in this area, and will not have much to add here. However, I did have a question about the logic in the paragraph beginning on line 56. If SCT predicts that people with different levels of SES will develop different cognitive styles, how would manipulating subjective SES in a one-time experiment impact those? Presumably stable cognitive styles are built up over a lifetime, so why would they be impact by a simple experimental manipulation?
Response: Thank you for your comments and valuable suggestions. By reviewing previous studies, we have obtained some research support. For example, wearing business suits induces individuals to exhibit psychological characteristics that closely resemble those who have spent extended periods in upper-class environments (M. W. Kraus & Mendes, 2014). In addition, researchers have utilized the cycles of poverty among sugarcane farmers, where they experience poverty before harvest and relative affluence after, to explore the causal relationship between naturally occurring poverty states and cognitive function. The results indicate that the cognitive performance of the same farmer is significantly lower before harvest (during poverty) compared to after harvest (during affluence), a difference that cannot be explained by variations in time, nutrition, or work effort (Mani, Mullainathan, Shafir, & Zhao, 2013). Based on previous research, brief priming of subjective SES can elicit behavioral patterns similar to those exhibited by individuals with long-term exposure to a specific SES (Guo, Yang, Li, & Hu, 2015; Michael W Kraus, Tan, & Tannenbaum, 2013), making it feasible to manipulate subjective SES and study its causal relationship with pro-environmental behavior.
Comments 5: Similarly, I don't really know the theory of life history strategy, but can it really be significantly impacted by an intervention in a laboratory?
Response: Thank you for your comments and valuable suggestions. Your suggestion is invaluable, and we have thoroughly explored the relevant literature. Unfortunately, direct research on this topic is lacking. However, a review of past empirical studies provides some insights. For instance, researchers have observed that individuals nearing payday tend to prioritize immediate benefits in intertemporal monetary decisions, indicating higher levels of time discounting (Carvalho, Meier, & Wang, 2016). Experience with poverty leads individuals to favor immediate small incomes over potential higher future earnings, resulting in reduced investment in long-term gains (J. Haushofer & Fehr, 2014). Additionally, experiments manipulating negative income shocks, common among those with low SES, have shown that such shocks increase delay discounting rates, while positive income changes decrease them (Johannes Haushofer, Schunk, & Fehr, 2019). What’s more, from the theoretical perspective, Psychological Shift Model under the threat perspective highlights three primary ways in which low SES influences psychology: resource scarcity, environmental instability and unpredictability, and adverse childhood experiences (Jennifer Sheehy-Skeffington & Haushofer, 2014; Wang, Li, Yang, Hu, & Du, 2022). Firstly, individuals' assessments of their control over life outcomes diminish, shifting their focus from the future to the present, from distant to local concerns, from distant social relationships to close ones, and from hypothetical to concrete thinking. Secondly, perceived control weakens, prompting individuals to shift from pursuing long-term goals to short-term ones, from seeking rewards to avoiding threats, and from employing slow strategies to fast ones. The third aspect involves concentration of cognitive resources on tasks and stimuli to meet immediate urgent needs, leading to adaptive decision-making (J. Sheehy-Skeffington, 2020; Wang et al., 2022). Therefore, the feeling of scarcity triggered by comparisons with those at the highest hierarchy may induce psychological shifts in individuals to cope with threats, thus may foster temporary preference states for short-term behaviors that benefit themselves.
Comments 6: Line 43: I believe this should be cited in a conventional format.
Response: Thank you for your comments and valuable suggestions. We have made revisions based on your suggestions. The specific changes are as follows:
Throughout history, society has advocated a cultural value that expects those with abundant resources (ie., high SES individuals) to contribute more to the overall welfare (BAI Jie, 2021). However, whether individuals of high SES engage more in pro-environmental behaviors, as society anticipates, remains inconclusive.
Study 1:
Comments 7: Line 164: Why are the researchers looking for a medium effect size?
Response: Thank you for your comments and valuable suggestions. Our study employed t-tests for data analysis (Single-factor two-level between-subjects design), and thus, a moderate effect size (effect size d = 0.5) was chosen for the calculations (Faul, Erdfelder, Lang, & Buchner, 2007) The article used for calculating the effect size has been cited over 60,000 times. In this cited article, on Page 178, it explicitly states that if data analysis employs a t-test, the medium effect size of d =0.5 is applicable. Therefore, we used a medium effect size.
Comments 8: Line 167: There should be supplemental information about who tends to participate in this online platform. Without knowing who was likely to be in the sample, there's no way of knowing what groups these results will generalize to.
Response: Thank you for bringing this to our attention. In this study, we utilized the Credamo platform, a well-established tool commonly used in authoritative journals for recruiting participants (Wei, Yu, Peng, & Zhong, 2023). Our study sample comprised individuals from the non-student population, and we conducted additional analyses to gather information on participants' occupations and educational backgrounds. Below are the detailed findings:
Table 1. The occupational distribution of participants across different studies
|
Educational levels |
Study 1 |
Study 2 |
Study 3 |
|
High school/secondary vocational school/technical school |
9 |
3 |
9 |
|
Junior college/night school/radio and television university |
15 |
9 |
17 |
|
Undergraduate |
187 |
106 |
269 |
|
Master's degree or above |
46 |
28 |
56 |
|
Occupational categories |
Study 1 |
Study 2 |
Study 3 |
|
Unemployed, semi-employed, or underemployed in urban or rural areas |
2 |
1 |
4 |
|
Non-technical workers or manual laborers, such as industrial workers and agricultural laborers |
48 |
22 |
40 |
|
Employees in commercial service industries, such as chefs, drivers, and hairdressers |
20 |
8 |
23 |
|
Self-employed workers or business owners |
47 |
30 |
62 |
|
Lower-middle class |
94 |
56 |
163 |
|
Upper-middle class |
39 |
27 |
51 |
|
Upper-class |
7 |
2 |
8 |
Table 2. The educational levels of participants across different studies
Comments 9: Line 170: The analyses should also be run including the excluded participants (at least the ones with extreme scores) as a robustness check.
Response: Thank you for your comments and valuable suggestions. We have revised the content based on your suggestion. We greatly appreciate your valuable suggestions. Here are the specific changes we made:
Additionally, when data within three standard deviations were not excluded, the results revealed a significant difference between the high ((M = 5.86, SD = 0.60)) and low SES priming conditions (M = 5.60, SD = 0.82), t (257) = −2.94, p < 0.01, Cohen's d = 0.36, demonstrating the robustness of the results.
Comments 10: Line 223: The claim that this validated the causal relationship of SES and pro-environmental behavior is too strong. The researchers have demonstrated a causal relationship between subjective SES and expressed intention of pro-environmental behavior, but this is a slightly weaker claim.
Response: Thank you for your comments and valuable suggestions. We have revised the statements based on your suggestions to make our manuscript more precise. We greatly appreciate your valuable suggestions (See lines 182-183).
Comments 11: Line 227: I don't see the need for this regression since the experimental design already rules out sex, age, and objective SES as reasons for the association. There's nothing wrong with including it, though. However, I think the researchers could use an instrumental variables approach to detect the causal effect of manipulating subjective SES on the outcome. Given the way the model is currently specified, I don't see how this can support any causal claims, since subjective SES varies for reasons that are not related to the experiment. The researchers claim that subjective SES "influences" the outcome is not supported by this model.
Response: Thank you for bringing this to our attention. Your suggestion is highly valuable to us. In fact, the causal relationship between subjective SES and preferences for green consumption has been demonstrated in the t-test results. The regression model was initially intended to reassess whether this relationship persists after controlling for objective SES, age, and gender (although we employed pseudo-randomization during participant recruitment to minimize these factors' effects). We have acknowledged that the regression model cannot establish causal relationships between variables, and as such, we have revised the statement accordingly. Your insightful suggestions are greatly appreciated. If you still feel the model might not add value, please let us know, and we can consider removing this section.
In addition, we have further studied the causal mediation models (Ge, 2023), which have been crucial for informing our future experimental designs on mediation. Thank you once again for your valuable suggestions.
Study 2:
Similar comments to before. In this case researchers DO demonstrate that manipulating subjective SES affects pro-environmental behavior.
Comments 12: Line 285: This study does not demonstrate that "individuals from high SES donate more money to environmental organizations, whereas those from low SES donate less". Rather, it demonstrates that people induced to have a higher subjective SES tend to donate more.
Response: Thank you for highlighting this. Based on your suggestion, we have revised the statement to emphasize that individuals induced with high subjective SES donated more to environmental organizations compared to those induced with low subjective SES. We sincerely appreciate your valuable suggestion (See lines 249-250).
Comments 13: Line 289: It may be true that "low SES individuals prefer instant gratification and are less willing to engage in environmentally friendly behavior for long-term development", but this study isn't enough to make this claim. In particular, I don't think the authors can claim that people assigned to the low subjective SES group are prioritizing instant gratification, since it's not known what they did with the money they chose not to donate.
Response: Thank you for your comments and valuable suggestions. In the concept of pro-environmental behavior: pro-environmental behavior refers to actions taken by individuals that benefit the environment, involving a trade-off between short-term self-interest and long-term environmental benefits (Barclay & Barker, 2020; Lange, Steinke, & Dewitte, 2018; M. Li et al., 2023; Mei Li et al., 2021; Niu, Fan, Ren, Li, & Zhong, 2023; Yi-Beng et al., 2022), at the expense of self-interests (Kollock, 1998; Steg & Vlek, 2009). For example, donating to sustainable environmental causes requires cash payments, whereas non-environmentally friendly behaviors do not require sacrificing immediate self-interests (e.g., cash). The smaller donation amounts indicate lower pro-environmental behavior among participants (Cakanlar, Nikolova, & Nenkov, 2022; Tam, 2022). In addition, researchers propose that when individuals consider the long-term benefits to nature (e.g., donating to environmental causes to promote sustainability), they need to suppress their instinctual desires (e.g., saving money), and that higher levels of self-control make individuals more likely to restrain their selfish impulses and exhibit pro-environmental preferences (Chuang et al., 2016). Based on these, we infer that individuals form low-SES donate less to environmental organizations due to prioritize short-term self- interests.
Thank you for your suggestions, we have incorporated the concept of pro-environmental behavior in the introduction. Additionally, we have acknowledged the limitation in our study: There is a possibility that the smaller donation amounts from individuals with low subjective SES may allocate the remaining funds to other types of pro-environmental behaviors post-experiment. Future research could design dedicated studies to explore this issue more comprehensively.
Study 3:
Comments 14: I'm curious why the authors didn't use a structural equations model, given that sense of control and life history strategy are both latent variables (and, of course, the outcome). Also, why did this study use the experimental manipulation? Study 2 already demonstrated that the manipulation impacts giving, and the study design does NOT allow the authors to claim that the mediation is causal (since subjective SES varies for reasons unrelated to the intervention), so I'm not clear why the experimental manipulation was used here.
Response: Thank you for your comments and valuable suggestions. We utilized Hayes' PROCESS macro 3.4 to analyze the chained mediation effect. This tool is favored by scholars for its ability to produce results similar to those from structural equation modeling (SEM), making it a widely adopted method in both domestic and international research studies (Cakanlar et al., 2022; Eom, Kim, & Sherman, 2018; Keqiang, Fenglan, Li, & Min, 2023; Liu & Li, 2019, 2022; Niu et al., 2023; ShijinShijin, 2021; Tanjitpiyanond, Jetten, & Peters, 2022).
Importantly, we apologize for any confusion regarding the experimental approach in Study 3 to investigate the influence of SES on environmental donation behavior. The reason for this design is partly based on references to previous literature. For example, Chuang et al. (2016) manipulated participants' self-construal and explored whether self-construal states influence green consumption preferences through self-control (trait variable). The results showed that induced self-construal indeed altered participants' self-control, thereby affecting subsequent preferences for green products. In addition, participants primed with power experienced a greater connection with their future selves compared to participants not primed with power, which mediated the temporal discounting rates (Joshi & Fast, 2013). In addition, we initially aimed to examine whether brief prime of SES could influence individuals' sense of control and life history strategies (psychological variables), potentially leading to behavioral changes (ie., pro-environmental behavior). Hence, Study 3 utilized the experimental approach rather than a questionnaire study.
We further analyzed the relationship between subjective SES, sense of control, and life history strategy to explore whether activating participants' subjective SES would change their current sense of control and life history strategy. The results are as follows:
4.2.4. Amount of sense of control and life history strategy
An independent samples t-test was conducted to examine subjective SES disparities in sense of control and life history strategy. The results revealed that participants in the high subjective SES condition (M = 5.13, SD = 0.93) had higher sense of control scores compared to those in the low subjective SES condition (M = 4.18, SD = 1.29), indicating a stronger sense of control (t(349) = 7.96, p < 0.001, Cohen’s d = 0.84). Additionally, participants in the high subjective SES condition (M = 5.68, SD = 0.46) had higher life history strategy scores compared to those in the low subjective SES condition (M = 5.44, SD = 0.50), indicating a tendency to adopt slower life history strategies (t(349) = 4.71, p < 0.001, Cohen’s d = 0.50).
Furthermore, we have revised the statements in the chain mediation model to emphasize that the model results reflect correlations rather than causal relationships. Thank you once again for your suggestions.
Comments 15: I think the authors should make more of the fact that life history strategy was not related to subjective SES. On the one hand, this is good since an experimental manipulation shouldn't be able to impact life history strategy. On the other hand, subjective SES varies for reasons that are not related to the intervention, and it seems surprising to me that these variables would not be associated, especially given the introduction of the paper where the authors claimed that SES should be related to life history strategies (and not only due to differences in sense of control). At the least this should be discussed.
Response: Thank you for your comments and valuable suggestions. Based on your valuable suggestions, we further analyzed the relationship between subjective SES, sense of control, and life history strategy to explore whether induced participants' subjective SES would change current psychological states of sense of control and life history strategy (See lines 321-330).
In addition, As the expert pointed out, the mediation analysis did not establish the pathway between subjective SES and the amount of environmental donations through the mediation of life history theory. We have now included the relevant discussion in the revised manuscript.
Discussion
Comments 16: Line 424: I don't see how a simple intervention manipulating subjective SES could be related to "form[ing] self-concepts and social cognitive tendencies based on individual attributes". Similarly, does life history theory allow for changes in strategies related to extremely small interventions like the one used in this study?
Response: Thank you for your comments and valuable suggestions.
Regarding the question raised in comment 4 about whether the manipulation of subjective SES is related to forming self-concepts and social cognitive tendencies: By reviewing previous studies, we have obtained some research support. For example, wearing business suits induces individuals to exhibit psychological characteristics that closely resemble those who have spent extended periods in upper-class environments (M. W. Kraus & Mendes, 2014). In addition, researchers have utilized the cycles of poverty among sugarcane farmers, where they experience poverty before harvest and relative affluence after, to explore the causal relationship between naturally occurring poverty states and cognitive function. The results indicate that the cognitive performance of the same farmer is significantly lower before harvest (during poverty) compared to after harvest (during affluence), a difference that cannot be explained by variations in time, nutrition, or work effort (Mani et al., 2013). Based on previous research, brief priming of subjective SES can elicit behavioral patterns similar to those exhibited by individuals with long-term exposure to a specific SES (Guo et al., 2015; Michael W Kraus et al., 2013), making it feasible to manipulate subjective SES and study its causal relationship with pro-environmental behavior.
Regarding the question raised in comment 5 about whether manipulating subjective SES influences life history strategies: Your suggestion is invaluable, and we have thoroughly explored the relevant literature. Unfortunately, direct research on this topic is lacking. However, a review of past empirical studies provides some insights. For instance, researchers have observed that individuals nearing payday tend to prioritize immediate benefits in intertemporal monetary decisions, indicating higher levels of time discounting (Carvalho et al., 2016). Experience with poverty leads individuals to favor immediate small incomes over potential higher future earnings, resulting in reduced investment in long-term gains (J. Haushofer & Fehr, 2014). Additionally, experiments manipulating negative income shocks, common among those with low SES, have shown that such shocks increase delay discounting rates, while positive income changes decrease them (Johannes Haushofer et al., 2019). What’s more, from the theoretical perspective, Psychological Shift Model under the threat perspective highlights three primary ways in which low SES influences psychology: resource scarcity, environmental instability and unpredictability, and adverse childhood experiences (Jennifer Sheehy-Skeffington & Haushofer, 2014; Wang et al., 2022). Firstly, individuals' assessments of their control over life outcomes diminish, shifting their focus from the future to the present, from distant to local concerns, from distant social relationships to close ones, and from hypothetical to concrete thinking. Secondly, perceived control weakens, prompting individuals to shift from pursuing long-term goals to short-term ones, from seeking rewards to avoiding threats, and from employing slow strategies to fast ones. The third aspect involves concentration of cognitive resources on tasks and stimuli to meet immediate urgent needs, leading to adaptive decision-making (J. Sheehy-Skeffington, 2020; Wang et al., 2022). Therefore, the feeling of scarcity triggered by comparisons with those at the highest hierarchy may induce psychological shifts in individuals to cope with threats, thus may foster temporary preference states for short-term behaviors that benefit themselves.
References
BAI Jie, Y. S., XU Buxiao, GUO Yongyu. (2021). How can successful people share their goodness with the world: The psychological mechanism underlying the upper social classes’ redistributive preferences and the role of humility. Acta Psychologica Sinica, 53(10), 1161
Barclay, P., & Barker, J. L. (2020). Greener than thou: people who protect the environment are more cooperative, compete to be environmental, and benefit from reputation. Journal of Environmental Psychology, 72, 101441.
Cakanlar, A., Nikolova, H., & Nenkov, G. Y. (2022). I Will Be Green for Us: When Consumers Compensate for Their Partners’ Unsustainable Behavior. Journal of Marketing Research, 60(1), 110-129.
Carvalho, L. S., Meier, S., & Wang, S. W. (2016). Poverty and Economic Decision-Making: Evidence from Changes in Financial Resources at Payday. Am Econ Rev, 106(2), 260-284.
Chuang, Y., Xie, X., & Liu, C. (2016). Interdependent orientations increase pro-environmental preferences when facing self-interest conflicts: The mediating role of self-control. Journal of Environmental Psychology, 46, 96-105.
Eom, K., Kim, H. S., & Sherman, D. K. (2018). Social class, control, and action: Socioeconomic status differences in antecedents of support for pro-environmental action. Journal of Experimental Social Psychology, 77, 60-75.
Faul, F., Erdfelder, E., Lang, A.-G., & Buchner, A. (2007). G* Power 3: A flexible statistical power analysis program for the social, behavioral, and biomedical sciences. Behavior research methods, 39(2), 175-191.
Ge, X. (2023). Experimentally manipulating mediating processes: Why and how to examine mediation using statistical moderation analyses. Journal of Experimental Social Psychology, 109.
Guo, Y., Yang, S., Li, J., & Hu, X. (2015). Social Fairness Researches in Perspectives of Social Class Psychology. Advances in Psychological Science, 23(8).
Haushofer, J., & Fehr, E. (2014). On the psychology of poverty. Science, 344(6186), 862-867.
Haushofer, J., Schunk, D., & Fehr, E. (2019). Negative income shocks increase discount rates.
Joshi, P. D., & Fast, N. J. (2013). Power and reduced temporal discounting. Psychol Sci, 24(4), 432-438.
Keqiang, M., Fenglan, L., Li, W., & Min, C. (2023). Childhood Socioeconomic Status and Mental Health of Rural Adult Residents: The Roles of Hope and Subjective Well-Being. journal of psychological science, 46(5), 1148.
Kollock, P. (1998). Social dilemmas: The anatomy of cooperation. Annual review of sociology, 24(1), 183-214.
Kraus, M. W., & Mendes, W. B. (2014). Sartorial symbols of social class elicit class-consistent behavioral and physiological responses: a dyadic approach. J Exp Psychol Gen, 143(6), 2330-2340.
Kraus, M. W., Tan, J. J., & Tannenbaum, M. B. (2013). The social ladder: A rank-based perspective on social class. Psychological Inquiry, 24(2), 81-96.
Lange, F., Steinke, A., & Dewitte, S. (2018). The Pro-Environmental Behavior Task: A laboratory measure of actual pro-environmental behavior. Journal of Environmental Psychology, 56(APR.), 46-54.
Li, M., Li, J., Zhang, G. F., Fan, W., Li, H., & Zhong, Y. P. (2023). Social distance modulates the influence of social observation on pro-environmental behavior: An event-related potential (ERP) study. Biological Psychology, 178, 8.
Li, M., Tan, M., Wang, S., Li, J., Zhang, G., & Zhong, Y. (2021). The effect of preceding self-control on green consumption behavior: the moderating role of moral elevation. Psychology Research and Behavior Management, 2169-2180.
Liu, J., & Li, H. (2019). How state anxiety influences time perception: moderated mediating effect of cognitive appraisal and attentional bias. Acta Psychologica Sinica.
LIU, J., & LI, H. (2022). How state anxiety influences retrospective time duration judgment: moderated mediating effect of cognitive appraisal and memory bias. Acta Psychologica Sinica, 54(12), 1455.
Mani, A., Mullainathan, S., Shafir, E., & Zhao, J. (2013). Poverty impedes cognitive function. Science, 341(6149), 976-980.
Niu, N., Fan, W., Ren, M., Li, M., & Zhong, Y. (2023). The Role of Social Norms and Personal Costs on Pro-Environmental Behavior: The Mediating Role of Personal Norms. Psychology Research and Behavior Management, 2059-2069.
Sheehy-Skeffington, J. (2020). The effects of low socioeconomic status on decision-making processes. Curr Opin Psychol, 33, 183-188.
Sheehy-Skeffington, J., & Haushofer, J. (2014). The behavioural economics of poverty. Barriers to and opportunities for poverty reduction, 96-112.
ShijinShijin, Y.-Z. K. S. (2021). Childhood socioeconomic status, life history strategy and consumptions: China traditional values of “unity and harmony” as moderator. journal of psychological science(1), 126.
Steg, L., & Vlek, C. (2009). Encouraging pro-environmental behaviour: An integrative review and research agenda. Journal of Environmental Psychology, 29(3), 309-317.
Tam, K.-P. (2022). Gratitude to nature: Presenting a theory of its conceptualization, measurement, and effects on pro-environmental behavior. Journal of Environmental Psychology, 79.
Tanjitpiyanond, P., Jetten, J., & Peters, K. (2022). How economic inequality shapes social class stereotyping. Journal of Experimental Social Psychology, 98.
Wang, T., Li, L., Yang, J., Hu, X., & Du, T. (2022). Low Socioeconomic Status and intertemporal choice: the mechanism of “psychological-shift” from the perspective of threat. Advances in Psychological Science, 30(8).
Wei, X., Yu, F., Peng, K., & Zhong, N. (2023). Psychological richness increases behavioral intention to protect the environment. Acta Psychologica Sinica, 55(8), 1330.
Yi-Beng, Z., Mei, L., Jin, L., Min, T., Wei, F., & Hui-E, L. (2022). “Pursue Reputation for Profit”: The Influence of social observation and social distance on the pro-environmental behavior. Journal of Psychological Science, 45(6), 1398.

Round 2
Reviewer 1 Report
Comments and Suggestions for Authors
Everything is OK now